# Desalination Using the Capacitive Deionization Technology with Graphite/AC Electrodes: Effect of the Flow Rate and Electrode Thickness

**DOI:** 10.3390/membranes12070717

**Published:** 2022-07-20

**Authors:** Jhonatan Martinez, Martín Colán, Ronald Catillón, Jesús Huamán, Robert Paria, Luis Sánchez, Juan M. Rodríguez

**Affiliations:** Center for the Development of Advanced Materials and Nanotechnology, Universidad Nacional de Ingeniería, Av. Túpac Amaru 210, Lima 15333, Peru; jhonatan.martinez.ore@gmail.com (J.M.); mcolan@niprojects.com.pe (M.C.); rcastillonh@uni.pe (R.C.); jesus.huaman.c@uni.pe (J.H.); rparias@uni.pe (R.P.); lasr_uni@hotmail.com (L.S.)

**Keywords:** activated carbon, capacitive deionization, CDI, water desalinization, graphite electrodes, porous materials, salt absorption capacity, specific energy consumption, cell design, cell Performance

## Abstract

Capacitive deionization (CDI) is an emerging water desalination technology whose principle lies in ion electrosorption at the surface of a pair of electrically charged electrodes. The aim of this study was to obtain the best performance of a CDI cell made of activated carbon as the active material for water desalination. In this work, electrodes of different active layer thicknesses were fabricated from a slurry of activated carbon deposited on graphite sheets. The as-prepared electrodes were characterized by cyclic voltammetry, and their physical properties were also studied using SEM and DRX. A CDI cell was fabricated with nine pairs of electrodes with the highest specific capacitance. The effect of the flow rate on the electrochemical performance of the CDI cell operating in charge–discharge electrochemical cycling was analyzed. We obtained a specific absorption capacity (SAC) of 10.2 mg/g and a specific energetic consumption (SEC) of 217.8 Wh/m^3^ at a flow rate of 55 mL/min. These results were contrasted with those available in the literature; in addition, other parameters such as Neff and SAR, which are necessary for the characterization and optimal operating conditions of the CDI cell, were analyzed. The findings from this study lay the groundwork for future research and increase the existing knowledge on CDI based on activated carbon electrodes.

## 1. Introduction

The continuous growth of the world population and the limited access to drinkable water resources have boosted the need to develop devices with the capacity to remove ions from a saline water solution [1]. In fact, the lack of drinkable water comes together with the lack of energy resources and human technical capacities. In that sense, it is necessary to propose new devices and techniques that take into account concepts of energy efficiency [2] and robustness. Capacitive deionization (CDI) is a technology that allows us to remove ions from a solution. It is usually composed of a CDI cell made of several pairs of electrodes electrically charged and uncharged at a specific frequency. The basic principle of CDI lies in the storage of ions (adsorption) when an external direct voltage, usually below 2.0 V, is applied to a system containing a saline solution; the electrically induced charged ions in the solution are adsorbed into the surface of the polarized electrodes, and as soon as the external electric field is reversed, the adsorbed ions are desorbed and fed back to a bulk solution, leading to a consequent regeneration of saturated electrode materials [3]. The efficiency of this process depends on the active material used to build the electrode. In order to optimize it, several factors have been pointed out, such as the absorption capacity, porosity, charge efficiency, diffuse charge and electrocapillary [4,5,6,7,8,9]. These kinds of factors directly affect each charge–discharge cycle [10]. For these reasons, new CDI cells are designed in order to maximize their performance, changing their active material porosity [11], thickness [12] and the geometry disposition of electrodes maximizing their active surface [13,14]. The materials examined for CDI electrodes are generally chosen based on features or characteristics such as high hydrophilicity, high porosity, high surface area and good electronic conductivity, and activated carbon (AC) meets all these criteria [15]. In this work, we fabricated a CDI cell based on electrodes made of activated carbon as the active material. The effect of the thickness of the active layer and flow rate on the electrochemical performance of the CDI cell was analyzed. In order to compare with other devices, we used the salt adsorption capacity (SAC) and the specific energy consumption (SEC); these parameters allowed us to standardize the characteristics of cells reported in other works [16].

## 2. Materials and Methods

### 2.1. Materials

Activated carbon (CAS 7440-44-0, M_w_~12.01, Sigma-Aldrich Co., Burlington, MA, USA), polyvinyl alcohol (CAS 9002-89-5 Mowiol, 10–98/M_w_~61,000, Sigma-Aldrich Co., Burlington, MA, USA), and glutaric alcohol (CAS 110-94-1, M_w_~132.11, Merck, Burlington, MA, USA) were used to prepare the active material. Commercial graphite sheets were used as the substrate. Sodium chloride (NaCl anhydrous, ≥99% purity, M_w_~58.44.11, Sigma-Aldrich, Burlington, MA, USA) was used to prepare the saline solution for desalination assays and the electrolytic solution for voltammetry cycle measurements.

### 2.2. Fabrication of the Electrode

To fabricate the electrodes, first, a slurry of the active material was prepared by using commercial activated carbon powder. The slurry was prepared as a suspension of activated carbon powder (45 g), polyvinyl alcohol (9 g) and glutaric alcohol (15 g) in 150 mL of water. The mixture was stirred for 2 h and sonicated for 40 min to ensure homogeneity. Second, the slurry was deposited on a graphite sheet by using the doctor blade method with the help of a film coater (MIT MSK-AFA-III-HB) and a Micrometer Adjustable Film Applicator (MIT EQ-Se-KTQ-100), which allowed us to modify the initial thickness of the wet film. Finally, the coated graphite sheets were dried first at room temperature for 2 h and then placed into an oven at 120 °C for one hour.

### 2.3. Fabrication of CDI Cell

The CDI cell consists of nine pairs of electrodes placed as is illustrated in Figure 1a. The electrode consists of two elementary components, the substrate, which in this case is a graphite sheet of 120 mm × 120 mm × 1 mm dimensions, and the active material, which covers an area of 100 mm × 100 mm. A blind rubber gasket of 120 mm × 120 mm × 1 mm dimensions, with a 100 mm × 100 mm central aperture through which the saline solution can circulate, was used to separate the electrodes. The pairs of electrodes were arranged in such a way that the solution circulating inside them flowed through all of them.

### 2.4. Electrochemical Setup

Figure 1b shows a schematic of the complete CDI system used for the electrochemical measurements; it consists of a 15 L container used to store a NaCl solution of 0.2 M, which was pumped into the CDI cell by using a peristaltic pump, model MasterFlex L/S Digital Dispensing Pump Drives; a 1.2 V DC source (PellTron 3005D) was used to energize the cell and a pH meter HANA HI 5522 was used to measure the ion concentration.

### 2.5. Characterization Techniques

After the drying process and without any further treatment, the obtained activated carbon-based electrodes were characterized by using scanning electron microscopy (SEM) to analyze the morphology of the samples (SEM Zeiss EVO-10). The structural properties were studied by using X-ray diffraction (diffractometer DRX Bruker D8 Advance). The electrochemical properties of the as-prepared electrodes were examined by using cyclic voltammetry (CV). CV tests were performed using a three-electrode system, where a carbon electrode deposited on a graphite sheet as support with an exposed surface area of 1 cm^2^ was made to have contact with the electrolyte (0.5 M NaCl solution), while a platinum rod and a saturated Ag/AgCl served as counter and reference electrodes, respectively. Voltammetry measurements were performed with a potentiostat (Metrohm Autolab PGSTAT204) at an operating window from 0.0 to 1.2 V vs. ref. (0.1 V) in a 0.5 M NaCl electrolyte. The specific capacitance C (F/g) was determined considering the mass (g) of the active material on the electrode surface (1 cm^2^) using Equation (1) (Hainan Wang et al. [17,18,19,20,21]): (1)$$C_{s}=\frac{\int \;idV}{2v\;\Delta V\;m}$$
where *C_s_* is the specific capacitance (F/g), *i* is the measured electric current at different voltages (A), *V* is the applied voltage (V), *v* is the scan rate (V/s) and *m* is the mass of active material on the electrode (mg).

### 2.6. CDI Measurements

The CDI setup described in Section 2.4 was used to characterize the desalination performance. Ion adsorption and desorption steps were carried out using constant potential mode at 1.2 V. The electrode regeneration was accomplished by reversing the applied voltage to 1.2 V. All experiments were carried out with an initial concentration of 0.2 M NaCl solution and a 15 L electrolyte tank. The influence of operation conditions and initial electrode thickness on the performance of the CDI system was investigated. Different electrode thicknesses (100, 200, 300, 400, 500 and 600 μm), and flow rates from 15 to 105 mL min^−1^ with steps of 15 mL min^−1^ were analyzed. Salt adsorption capacity (SAC) and salt adsorption rate (SAR) were calculated from recorded ion concentration data. SAC and SAR were used to describe the salt adsorption capacity and rate of salt adsorption since such nomenclature is commonly used in the CDI community. 

## 3. Results and Discussions

In order to get the most efficient AC electrode, we fabricated and characterized electrodes with different initial thicknesses of the wet film deposited on a graphite sheet. Figure 2 shows SEM images of the top view of the electrodes with (a) 200 μm, (b) 400 μm and (c) 600 μm initial thicknesses. The images were obtained to visualize the morphology of the electrodes and also to verify if the increase of thicknesses would have some effect on the morphology of AC electrodes. There was no significant change in the morphology of the electrodes, which was hardly differentiable between them, as shown in Figure 2a–c. The final thickness of the electrodes, after the drying process, was measured from SEM images of the cross-section of the electrodes using the software ImageJ. Results are shown in Table 1 and Figure 2d. It should be noted that the initial thickness of the active layer (with water content) was set up by the EQ-Se-KTQ-100 micrometer. Figure 3a shows the variation of the measured thickness of the electrode as a function of the initial thickness of the activated carbon films with water content. Figure 4 shows SEM images of cross-section and histograms showing the distribution of measured thickness of electrodes of an initial thickness of the wet film of (a) 200 μm, (b) 400 μm and (c) 600 μm. The analysis of these results shows a small but significant reduction in thickness after the drying process.

Diffraction patterns of the electrodes of an initial thickness of 200 μm, 400 μm and 600 μm of active layer on a graphite sheet and after the drying process are shown in Figure 3b; all diffraction patterns show a strong diffraction peak at 2θ = 26.6° and a small peak at 2θ = 54.6°. Those peaks correspond to (002) and (004) crystalline planes of pure graphite [22]. From the same diffractograms, it is possible to observe the attenuation of the intensity peaks as the thickness of the active layer increases. In order to study the amorphous or crystalline nature of the AC active layer, electrodes of 300 and 600 μm of initial thickness deposited on SiO_2_ were fabricated following the same procedure as was described before. The inset in Figure 3b shows diffraction patterns of this electrode. The diffractograms show the amorphous nature of the material of the active layer. The diffractograms have two prominent, broad asymmetric peaks at 2θ of around 23° and 44°, which correspond to the planes (002) and (101), respectively, of amorphous carbon [23,24,25].

### 3.1. Electrochemical Characterization

Figure 5 shows the variation of specific capacitance as a function of measured thicknesses of activated carbon mixture deposited on graphite sheets (also shown in Table 1). Calculated by using the Equation (1) and Cyclic voltammograms show in Appendix A Figure A1. According to the graph, the specific capacitance grows until it reaches a maximum value; after that, it decreases. This behavior can be explained by the ion transportation theory, which states that the difficulty of transporting ions inside the active layer increases as the film thickness increases and due to an attenuation effect of the electric field due to the increase of active mass that covers the electrode, decreasing its ion absorption capacity. This is a critical point for the choice of electrode thickness [26]. A specific capacitance peak of 69.4 F/g was found in the graph, corresponding to a measured thickness of 102 μm. This result is consistent with the findings of Gamby [27], who predicted a maximum between 60 F/g and 115 F/g for this thickness. The specific capacitance is used to predict the electrochemical performance of an electrode; the higher its specific capacitance, the more ions it can retain in the charged double layer, and therefore a better efficiency can be obtained.

### 3.2. Cell Performance

In this section, the performance of a CDI cell will be analyzed. The CDI cell was fabricated using nine pairs of electrodes of the highest specific capacitance determined in the previous section, corresponding to an initial active layer thickness of 300 μm corresponding to a measured thickness of 102 μm. Figure 6a shows the ion concentration in ppm for multiple charge–discharge cycles. Three deionization cycles were counted in a 5000 s time interval. The graph depicts the typical shape of a charge–discharge for a CDI cell, which is similar to what has been observed elsewhere [10]. The observed cycles all have the same shape and present the same minimum and maximum values, resulting in equal cycle performances.

The salt removed in a cycle (*N_eff_*) determines the average amount of salt (mg) that the electrodes together can retain during the desalination process. It can be determined using Equation (2). Figure 6b shows how the *N_eff_* varies as a function of the input flow rate and the initial concentration of the electrolyte. The electrodes can hold a maximum value of 157 mg of salt from the electrolyte, which corresponds to a flow rate of 15 mL/min, for an initial electrolyte concentration of about 0.21 molar.
(2)$$N_{eff}=\int_{t_{c,0}}^{t_{c,f}}Q\;\left[C_{o}-C\left(t\right)\right]dt$$
where *Q* is the flow rate (mL/min), *Co* is the initial salt concentration (g/mL), *C*(*t*) is the time-varying salt concentration during adsorption (g/mL), *t*_c,0_ corresponds to the initial time in which the cell is energized and *t_c,f_* is the final time in which the system reaches equilibrium (min). Figure 6a details the time taken for each cycle during the data acquisition process. Based on 3 complete cycles (charge–discharge), the initial concentration (ppm) and lowest concentration (ppm) values were recorded for the respective calculations.

Salt removal rate (*SAR*) determines the value of the efficiency of the system expressed as a percentage; when a certain amount of salt is removed from an electrolytic solution during a cycle, it can be calculated according to Equation (3): (3)$$SAR=\frac{\int_{t_{c,0}}^{t_{c,f}}\;\left[C_{o}-C\left(t\right)\right]dt}{C_{o}\left(t_{c,f}-t_{c,0}\right)}$$

Figure 6b shows how the *SAR* varies depending on the inlet flow rate and the initial concentration of the electrolyte, with 14.1% being the highest percentage value that corresponds to the amount of salt retained in the desalination process with respect to the amount of salt that was initially in the volume of water. The maximum *SAC* was found for a flow rate of 15 mL/min, for an initial electrolyte concentration of about 0.21 molar. Although both parameters, *N_eff_* and *SAR*, show a noisy variation, the general trend is a decrease of both variables as the flow rate increases. 

Salt removal capacity (*SAC*) is defined as the capacity of the electrode to retain a certain amount of salt (mg) per unit mass of the electrode (g). In order to compare our results with those described in the literature, the *SAC* was calculated using Equation (4), where *Co* is the initial ion concentration, *C_D_* is the equilibrium ion concentration, *V* is the volume of electrolyte that goes through the CDI cell and *m_at_* is the total active mass of the electrodes:(4)$$SAC=\frac{\left(C_{0}-C_{D}\right)\cdot V}{m_{at}}$$

Figure 7 shows the variation of the SAC for different flow rates. The obtained *SAC* ranged from 7.5 to 10.2 mg/g. The highest value obtained is commonly chosen, but it is necessary to compare the time required for each charging period of the CDI cell since these data provide information on the ideal operating conditions of the cell. The maximum *SAC* value found was 10.2 mg/g for a flow rate of 55 mL/min. Table 2 shows the maximum *SAC* obtained and various reported results in the literature. It should be noticed that we only considered the higher reported values. Differences between materials and operational parameters (salt molarity, potential, active mass, active surface, electrode thickness, morphological structure, number of electrode pairs, etc.) can change each *SAC*. Table 2 compares the salt adsorption capacity of our CDI cell designed in this study against others found in the literature; for this purpose CDI cells based on activated carbon electrodes under similar operational parameters were selected. From the results shown in Table 2 we can point out that our CDI cell can operate with higher salt concentrations; thus, it could operate under more realistic conditions.

The average salt adsorption rate (*ASAR*), is defined as the capacity of absorption within a specific time, i.e., a time cycle or adsorption cycle time (*T_Ad_*), and was calculated using Equation (5):(5)$$ASAR=SAC/T_{Ad}$$

Figure 8 shows the *T_Ad_* as a function of the flow rate. It clearly shows that if the flow rate increases, the *T_Ad_* decreases. A large *T_Ad_* allows a better ion absorption to be obtained, resulting in a high *SAC*. However, a large *T_Ad_* produces a small *ASAR*, as shown in Figure 8.

### 3.3. Specific Energy Consumption

Specific energy consumption in CDI (*SEC*), represents the energy consumption per unit volume of water produced (Wh/m^3^) and is calculated using Equation (6): (6)$$SEC=\frac{1}{V_{D}}\int_{t_{c,0}}^{t_{d,f}}V_{cell}\left(t\right)\cdot i\left(t\right)dt$$

Figure 9a shows a typical current variation in cycles of absorption–desorption of the CDI cell. These data were acquired using multimeter UNI-T 71B true RMS and were used to calculate the *SEC*.

Figure 9b shows *T_Cyle_* and *SEC* variation as a function of the flow rate. *T_Cyle_* and *SEC* decrease as the flow rate increases. In this case, the *SEC* corresponding to a flow rate of 55 mL/min is 217.8 Wh/m^3^. 

## 4. Conclusions

In this work, the effects of the thickness of the active material and flow rate on the performance of a CDI cell, based on activated carbon electrodes, were studied. In order to obtain the most efficient AC electrode, we fabricated and characterized electrodes with different AC thicknesses deposited onto a graphite sheet. We found that electrodes with a thickness of 102 μm showed the best electrochemical properties, and therefore they were utilized to fabricate and analyze the performance of the CDI cell. We obtained a specific absorption capacity (SAC) of 10.2 mg/g and a specific energetic consumption (SEC) of 217.8 Wh/m^3^ at a flow rate of 55 mL/min. These results were contrasted with results available in the literature, finding that our developed CDI cell, operating with a higher initial salt feed concentration, showed a higher performance.

## Figures and Tables

**Figure 1 membranes-12-00717-f001:**
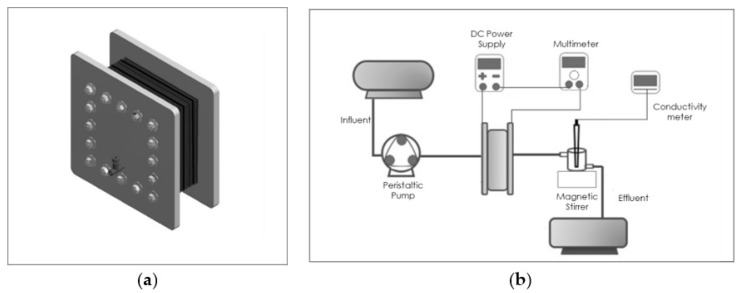
(**a**) CDI Cell structure with the electrode pairs in (**b**) schematic CDI system in which the different elements (CDI cell, peristaltic pump, DC power supply, conductivity and pH meter) are displayed.

**Figure 2 membranes-12-00717-f002:**
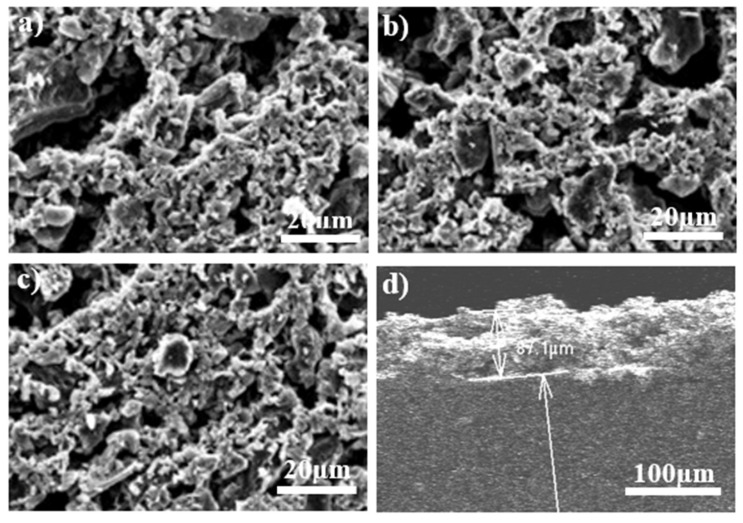
SEM images of top view of electrodes with (**a**) 200 μm, (**b**) 400 μm and (**c**) 600 μm initial thicknesses. (**d**) Cross-section view of an electrode of 200 μm initial thicknesses.

**Figure 3 membranes-12-00717-f003:**
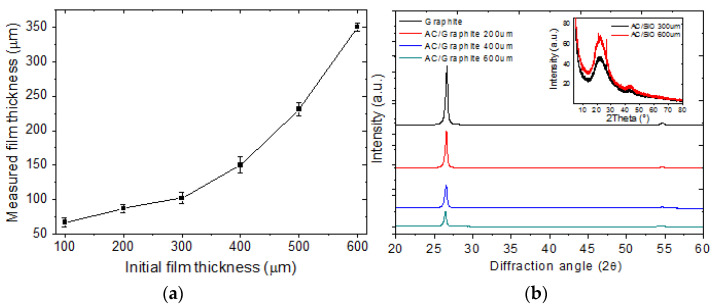
(**a**) Variation of measured film thickness as a function of initial film thickness. (**b**) DRX patterns for various electrode thicknesses.

**Figure 4 membranes-12-00717-f004:**
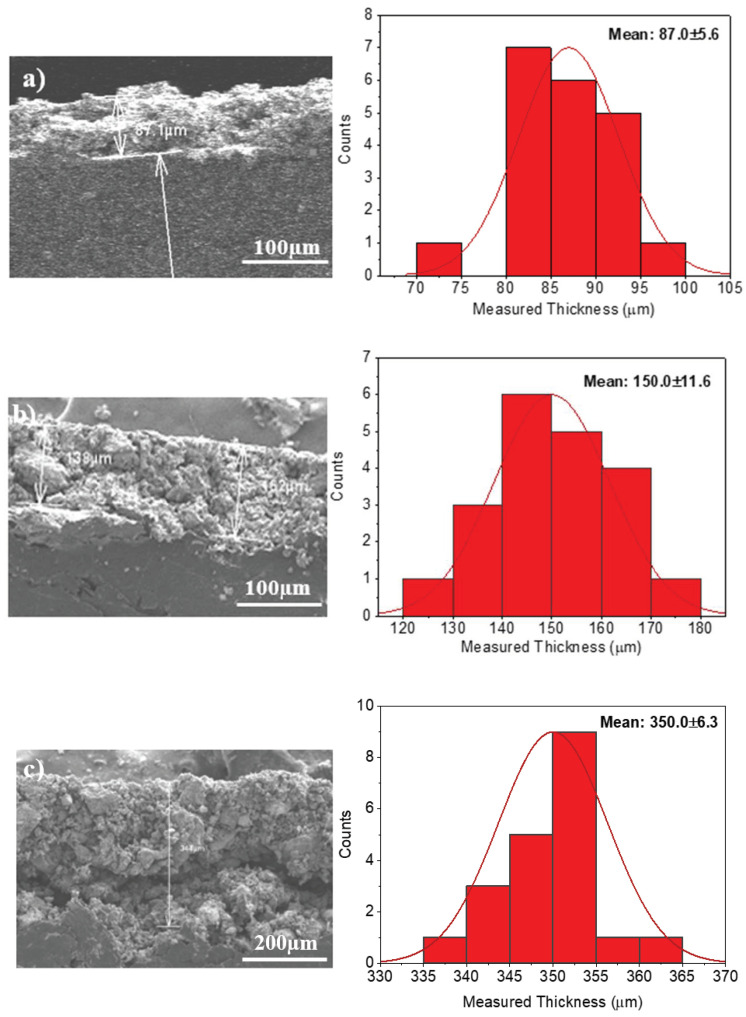
SEM images of cross-section and histograms showing the distribution of measured thickness of electrodes of an initial thickness of the wet film of (**a**) 200 μm, (**b**) 400 μm and (**c**) 600 μm.

**Figure 5 membranes-12-00717-f005:**
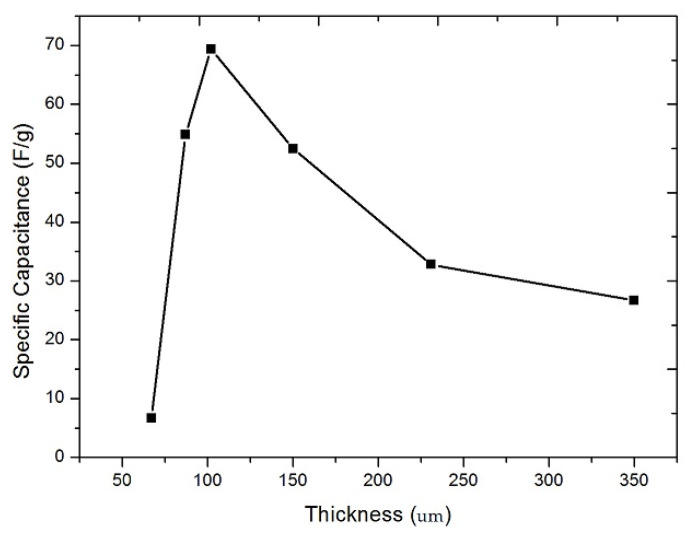
Specific capacitance for different thicknesses of activated carbon electrodes applied onto a graphite electrode.

**Figure 6 membranes-12-00717-f006:**
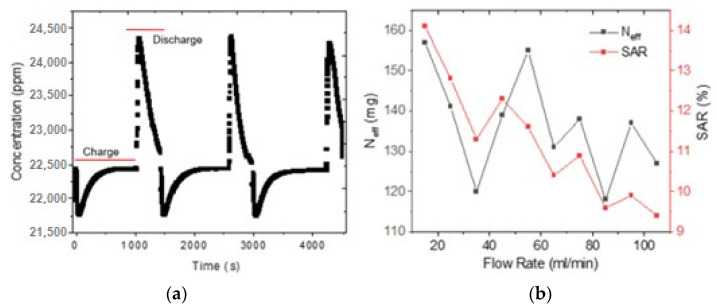
(**a**) Deionization cycles (charge–discharge) of CDI cell (NaCl 0.2 M, 1.2 V, 80 mL/min). (**b**) N_eff_ and SAR variation as a function of flow rate.

**Figure 7 membranes-12-00717-f007:**
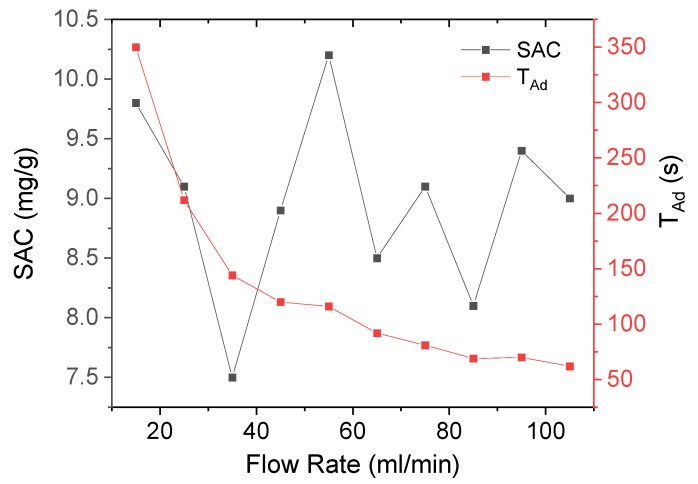
Variation of *SAC* and *T_AD_* for various flow rates (0.2 M, 1.2 V).

**Figure 8 membranes-12-00717-f008:**
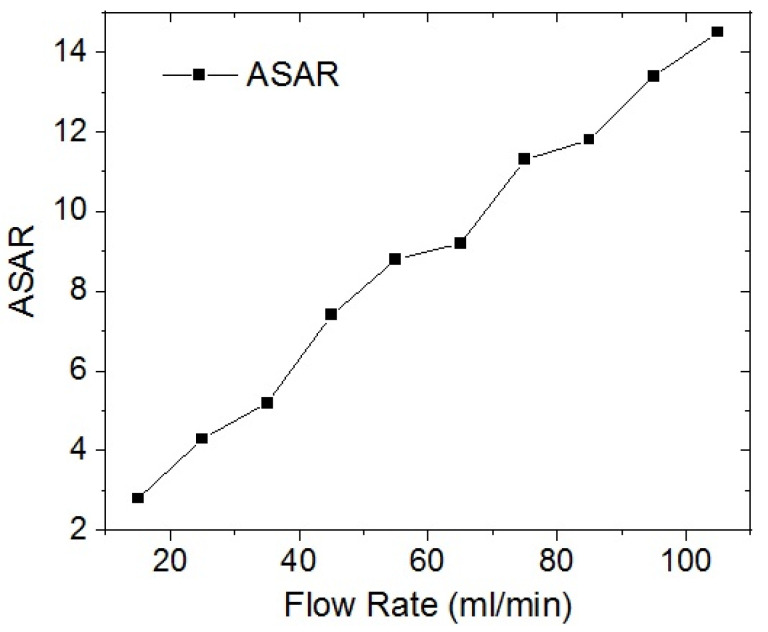
*ASAR* as a function of flow rate (0.4 M, 1.2 V).

**Figure 9 membranes-12-00717-f009:**
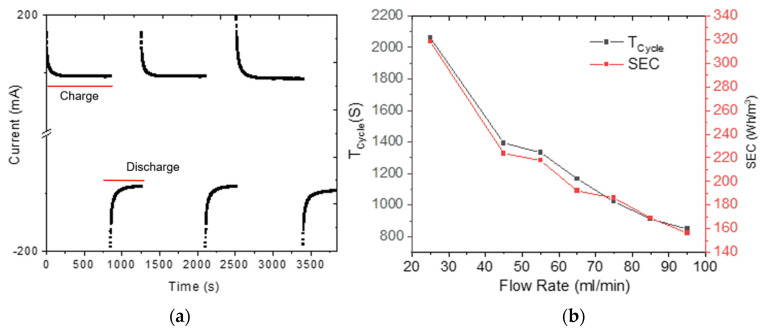
(**a**) A typical current variation of absorption-desorption cycles of the CDI (0.2 M, 1.2 V, 70 mL/min). (**b**) *T_Cyle_* an *SEC* variation as function of flow rate.

**Table 1 membranes-12-00717-t001:** Comparison between initial film thickness and measured thickness, specific capacitance and effective mass of the active material on the electrode for various electrode thicknesses.

Sample	Initial Film Thickness	Measured Thickness	Mass	Specific Capacitance
	(µm)	(µm)	(mg)	(F/g)
S1	100	67	8	6.7
S2	200	87	10	54.9
S3	300	102	12	69.4
S4	400	150	18	52.5
S5	500	231	28	32.8
S6	600	350	43	26.7

**Table 2 membranes-12-00717-t002:** Comparison of *SAC* reported with this work.

Type/Operation	*V*	Concentration (mM)	*SAC* (mg/g)
AC/CDI	1.2	35	8.7 [28]
AC/CDI	1.2	50	8.81 [29]
AC/CDI	1.2	58.4	2.7 [30]
AC/CDI	1.2	100	4.35 [31]
AC/CDI	1.2	100	4.92 [31]
AC/CDI	1.2	300	7.37 [31]
AC/CDI	1.2	200	10.2 (this work)

## Data Availability

Not applicable.

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
