# Peer review of "Desalination Using the Capacitive Deionization Technology with Graphite/AC Electrodes: Effect of the Flow Rate and Electrode Thickness"

_membranes, 2022, doi:10.3390/membranes12070717_

Round 1

Reviewer 1 Report

My comments are as follows,

1.  From equation 1 authors used the integral why did you use the integral?

2. What is the value of integral did you calculate the equation 1 you used 2 explain this? my questions belong in bold letters below

??=???2? Δ? ?

3. The XRD patterns from the inset figure 3 (b) 600 μm showed one shoulder peaks the 2theta around 29-30 ?

4. Also the inset figure 3 (b) XRD pattern figure and figure captions are not clear author changed them to use a clear one.

5. What is the mass ratio of the active working electrode.

6. Authors should improve the conclusion part.

7. If possible authors will add the CV figures to the revised manuscript.

Author Response

  1. From equation 1 authors used the integral why did you use the integral?

We used the equation (1) to calculate the specific capacitance based on multiple relevant research works found in the literature, some of them were added to the reference of our work.

Here below are the added references:

[17]. Daohua Liu, Xun-an Ning, Yanxiang Hong, Yang Li, Qiushi Bian, Jianpei Zhang. Covalent triazine-based frameworks as electrodes for high-performance membrane capacitive deionization. Electrochimica Acta, Volume 296, 2019, Pages 327-334. https://doi.org/10.1016/j.electacta.2018.10.044

[18]. H. A. Maddah, and M. A. Shihon, "Activated Carbon Cloth for Desalination of Brackish Water Using Capacitive Deionization", in Desalination and Water Treatment. London, United Kingdom: IntechOpen, 2018. https://www.intechopen.com/chapters/61156

[19]. Mingxing Shi, Hua Qiang, Chunyu Chen, Zahira Bano, Fengyun Wang, Mingzhu Xia, Wu Lei. Construction and evaluation of a novel three-electrode capacitive deionization system with high desalination performance. Separation and Purification Technology,Volume 273, 2021. https://doi.org/10.1016/j.seppur.2021.118976

[20]. Linlin Wu, Mingquan Liu, Silu Huo, Xiaogang Zang, Min Xu, Wei Ni, Zhiyu Yang, Yi-Ming Yan. Mold-casting prepared free-standing activated carbon electrodes for capacitive deionization. Carbon, Volume 149, 2019, Pages 627-636. https://doi.org/10.1016/j.carbon.2019.04.102

[21]. Hainan Wang, Laurent Pilon. Physical interpretation of cyclic voltammetry for measuring electric double layer capacitances. Electrochimica Acta, Volume 64, 2012, Pages 130-139. https://doi.org/10.1016/j.electacta.2011.12.118.

  1. What is the value of integral did you calculate the equation 1 you used 2 explain this? my questions belong in bold letters below

                      (1)

We used the equation (1) to calculate the specific capacitance based on multiple relevant research works found in the literature. It easy of demonstrate as follows:

                                                                  y   

Where: , is the charge and t is the time.

 Obs:     

For direct sweep:

For the reverse sweep:

Subtracting the forward and reverse sweep equations respectively.

  1. The XRD patterns from the inset figure 3 (b) 600 μm showed one shoulder peaks the 2theta around 29-30 ?

Thank you for pointing this out. We agree with this comment. Therefore, we modified the XRD analysis to include this point as follows:

Diffraction patterns of the electrodes of an initial thickness of 200 μm, 400 μm, 600 μm of active layer on a graphite sheet and after the dried process are shown in Figure 3 (b); All diffraction patterns show a strong diffraction peak at 2θ=26.6o and a small peak at 2θ=24.6o. Those peaks correspond to (002) and (004) crystalline planes of pure graphite [22]. From the same diffractograms, it is possible to observe the attenuation of the intensity peaks as the thickness of the active layer increases. In order to study the amorphous or crystalline nature of AC active layer, electrodes of 300 and 600 μm of initial thickness de-posited on SiO were fabricated flow the same procedure as was described before. The in-sect in figure 3 (b) shows diffraction patterns of this electrode. The diffractograms show the amorphous nature of the material of the active layer. The diffractograms have two prominent, broad asymmetric peaks at 2θ of around 23° and 44°, which correspond to the plane (002) and (101) respectively of amorphous carbon [23-25].

  1. Also the inset figure 3 (b) XRD pattern figure and figure captions are not clear author changed them to use a clear one.

As suggested by the reviewer. We improved the quality of XRD images.

  1. What is the mass ratio of the active working electrode.

In table 01 was included the active mass of the working electrode.

  1. Authors should improve the conclusion part.

We agree with the reviewer’s suggestion. The conclusion was modified as follow:

In this research work, the effects of the thickness of the active material and flow rate in the performance of a CDI cell based on activated carbon electrodes were studied. In or-der to get the most efficient AC electrode, we fabricated and characterized electrodes with different thicknesses deposited on a graphite sheet. We found that electrodes with a thick-ness of 102 μm possess the best electrochemical properties and therefore were utilized to fabricate our CDI cell. The performance of the CDI cell was analyzed. We obtained a spe-cific absorption capacity (SAC) of 10.2 mg/g, and a specific energetic consumption (SEC) of 217.8 Wh/m3 at a flow rate of 55 ml/min. These results were contrasted with results avail-able in the literature. We can highlight that our CDI cell operating with a higher initial salt feed concentration results in higher performance.

  1. If possible authors will add the CV figures to the revised manuscript.

We agree with the reviewer’s suggestion, thus some CV figures were added on the Supplementary Materials section of our manuscript.

Reviewer 2 Report

The article describes preparation of carbon electrodes and evaluation of CDI performance using 0.2 M NaCl and studies the effect of flow rates on salt removal efficiency and energy consumption. It is hard to comprehend the novelty of the work and the objective of the study since the electrode materials are very common and Capacitive deionization has been around for along time and several advances have already been reported. Authors have not cited any references in the last 3-5 years which describe a lot of improvements in the CDI technology. The presentation of the manuscript is also not up to the journal standard. I recommend that the manuscript be  reconsidered after some major changes are made. 

1. Overall, the manuscript needs a thorough English review as well as the improvement in quality of data and figures presented. 

2. Abstract - please also state the objectives of the study along with what advancements are being reported.

3. Lines 17-18. This statement is not grammatically correct. Please revise. 

4. Line 30 - Please remove semicolons as punctuation marks, where not necessary. 

5. Line 42-43, [8],[9]...Please reformat the reference citations according to the template. 

6. Line 49-50. please rephrase the "activated carbon-based as active material" to "activated carbon as the active material".

7. Line 91-92. Please elaborate on the sampling methods. Was there any coating of conductive materials before SEM imaging?

8. Lines 107-111. The desalting, cleaning and purging cycle durations are not mentioned anywhere. Also, the feed and the desalted feed conductivities are also not mentioned. 

9. Line 114. These flow rates do not include all the flow rates used in the experiments in Figure 5, 6, etc.

10. Lines 131-132. Please revise this statement The differences in the initial and measured thicknesses are small but significant. 

11. able 1. The mass and specific capacitance are functions of actual thickness and not the initial wet film thickness. Initial film thickness is a process variable and the measured thickness is the actual film property which affects the specific capacitance. Please include the measured thickness values for the remaining 3 samples or provide a valid justification of why the samples have been left out. 

12. Section 3.3 - it is unlikely that the electrode thickness, after reducing during drying, would return to the initial thickness values. It is not accurate to rely on the initial thickness as a parameter affecting the specific capacitance. The best capacitance value was found for the initial thickness value of 300 micrometer but the actual thickness is not known for this sample. Please measure the actual thickness for the electrodes to correlate with the capacitance value. 

13. Error bars needed for Figure 4 data.

14. Lines 176-177. Authors should specify that the Neff of 157 at 15 mL/min at the used feed concentrations. The effect of flow rate on the salt removal efficiency also depends on the feed concentration. 

15. Please label the charging and dis-charging cycles in the Figure 5(a).

16. Figure 5 caption. "Neff" should be changed to "Neff".

17. Figure 5(b), It is just mentioned that Neff and SAR vary with flow rate but no explanation is given for the random variation observed for both factors as flow rate increases. 

18. Line 188, "SAR varies depending on the inlet flow rate" . SAR also depends heavily on the feed concentration. Please revise.

19. Lines 195-196, specify the actual terms used in the equation, e.g. CD and not C, mat and not m.

20. Lines 200-201. "it shows a downward tendency....". This is not true. There is random variation in the SAC as the flow rate increases. Please provide a justification for this observation. 

21. Lines 208 - 210. From the Table2, it is obvious that the operational parameters referenced are not similar at all to those in this study. Please explain which similar operational parameters are being referred to? As you may know, the SAC valued depend on a lot of factors, not just the applied voltage. The feed concentration, electrode surface area, pore structure, feed flow rate, salt type, among many others. The feed concentrations in the references in Table 2 are vastly different and it is not accurate to draw a comparison with those studies without specifying the similar factors. Please revise the whole comparison and interpretation of the data presented in the Table2 and provide a reasonable explanation of how your electrodes compare to the literature.

21. Table 3, again, there is no mention of the operating conditions used in the references, which are similar to the current study. It is not scientific to club all the data in a simple table and draw a conclusion. SEC is also a function of lot of factors. Are all the factors similar in all the studies reported in Table 3? please explain. 

22. Why is the Figure A1 in the supplementary section? Please move it to main text and reference it with an explanation. 

23. Line 241. Where did the 16 W/m3 come from? The value is 217.8 Wh/m3 in Table 3. Please clarify.

24. The conclusion does not explain any correlation between the electrode material, properties and the CDI performance observed. There is very minimal in terms to addition to scientific body of knowledge in the area of capacitive deionization. 

25. There are two Figures number 6. Please review the whole manuscript and rearrange. 

26. References are not consistent. Please revise. 

Author Response

  1. Overall, the manuscript needs a thorough English review as well as the improvement in quality of data and figures presented.

It was reviewed and improved as the reviewer suggested

  1. Abstract - please also state the objectives of the study along with what advancements are being reported.

As suggested by the reviewer. We improved the abstract as follow:

Capacitive deionization (CDI) is an emerging water desalination technology whose principle lies in ion electrosorption at the surface of a pair of electrically charged electrodes. The main aim of this study is to get the best performance of a CDI cell made of activated carbon as the active material for water desalination. In this work, electrodes of different active layer thicknesses have been fabricated from a slurry of activated carbon deposited on graphite sheets. The as-prepared electrodes were characterized by cyclic voltammetry, and their physical properties were also studied using SEM and DRX. A CDI Cell was fabricated with nine pares of electrodes with the highest specific capacitance. The effect of the flow rate on the electrochemical performance of the CDI cell operating in charge-discharge electrochemical cycling was analyzed. We obtained a specific absorption capacity (SAC) of 10.2 mg/g, and a specific energetic consumption (SEC) of 217.8 Wh/m3 at a flow rate of 55 ml/min. These results were contrasted with results available in the literature, in addition, other parameters such as Neff and SAR were analyzed, which are necessary for the characterization and found the optimal operating conditions of the CDI cell. The findings from this study lay the groundwork for future research and increase the existing knowledge on CDI based on activated carbon electrodes.

  1. Lines 17-18. This statement is not grammatically correct. Please revise.

We modified that statement as follow:

We obtained a specific absorption capacity (SAC) of 10.2 mg/g, and a specific energetic consumption (SEC) of 217.8 Wh/m3 at a flow rate of 55 ml/min.

  1. Line 30 - Please remove semicolons as punctuation marks, where not necessary.

We removed the semicolons.

  1. Line 42-43, [8],[9]...Please reformat the reference citations according to the template.

We have fixed the error in line 45.

  1. Line 49-50. please rephrase the "activated carbon-based as active material" to "activated carbon as the active material".

It was rephrased in line 53.

  1. Line 91-92. Please elaborate on the sampling methods. Was there any coating of conductive materials before SEM imaging?

It was clarified (Line 94-95):

After the dry process and without any further treatment, the obtained activated carbon based electrodes were characterized by using …

  1. Lines 107-111. The desalting, cleaning and purging cycle durations are not mentioned anywhere. Also, the feed and the desalted feed conductivities are also not mentioned.

Initial conditions are mentioned in lines 117-120 and also in figure 6.

  1. Line 114. These flow rates do not include all the flow rates used in the experiments in Figure 5, 6, etc.

We included all flow rate analyzed in this study in line [1221-122]

  1. Lines 131-132. Please revise this statement The differences in the initial and measured thicknesses are small but significant.

We agree with this comment. Therefore, we modified that statement in line 139 as follows:

The analysis of these results shows a small but significant reduction in thickness after the dried process.

  1. Table 1. The mass and specific capacitance are functions of actual thickness and not the initial wet film thickness. Initial film thickness is a process variable and the measured thickness is the actual film property which affects the specific capacitance. Please include the measured thickness values for the remaining 3 samples or provide a valid justification of why the samples have been left out.

We totally agree with the reviewer’s comment. Therefore, the thicknesses missing in table 01 ware completed.

  1. Section 3.3 - it is unlikely that the electrode thickness, after reducing during drying, would return to the initial thickness values. It is not accurate to rely on the initial thickness as a parameter affecting the specific capacitance. The best capacitance value was found for the initial thickness value of 300 micrometer but the actual thickness is not known for this sample. Please measure the actual thickness for the electrodes to correlate with the capacitance value.

We agree with the reviewer’s comment. All thicknesses were measured, and we found that the actual thickness corresponding to the best capacitance was 102 μm. All this was clarified in the manuscript from line 167 to 180 lines.

  1. Error bars needed for Figure 4 data.

Specific capacitance was obtained using the area, so that we can’t calculate the error bar of each measure

  1. Lines 176-177. Authors should specify that the Neff of 157 at 15 mL/min at the used feed concentrations. The effect of flow rate on the salt removal efficiency also depends on the feed concentration.

  1. Please label the charging and dis-charging cycles in the Figure 5(a).

Labels of charging and discharging were added on figures 6a and 9a.

  1. Figure 5 caption. "Neff" should be changed to "Neff".

Fixed

We have fixed the error in line 206.

  1. Figure 5(b), It is just mentioned that Neff and SAR vary with flow rate but no explanation is given for the random variation observed for both factors as flow rate increases.

  1. Line 188, "SAR varies depending on the inlet flow rate" . SAR also depends heavily on the feed concentration. Please revise.

  1. Lines 195-196, specify the actual terms used in the equation, e.g. CD and not C, mat and not m.

We have fixed it in lines 221-222

  1. Lines 200-201. "it shows a downward tendency....". This is not true. There is random variation in the SAC as the flow rate increases. Please provide a justification for this observation.

Clarified

  1. Lines 208 - 210. From the Table2, it is obvious that the operational parameters referenced are not similar at all to those in this study. Please explain which similar operational parameters are being referred to? As you may know, the SAC valued depend on a lot of factors, not just the applied voltage. The feed concentration, electrode surface area, pore structure, feed flow rate, salt type, among many others. The feed concentrations in the references in Table 2 are vastly different and it is not accurate to draw a comparison with those studies without specifying the similar factors. Please revise the whole comparison and interpretation of the data presented in the Table2 and provide a reasonable explanation of how your electrodes compare to the literature.

Clarified

  1. Table 3, again, there is no mention of the operating conditions used in the references, which are similar to the current study. It is not scientific to club all the data in a simple table and draw a conclusion. SEC is also a function of lot of factors. Are all the factors similar in all the studies reported in Table 3? please explain.

We decided to not consider the table and erased it from the manuscript because on literature is hard to find SEC’s which have the same initial conditions, almost concentration and protentional

  1. Why is the Figure A1 in the supplementary section? Please move it to main text and reference it with an explanation.

Fixed

  1. Line 241. Where did the 16 W/m3 come from? The value is 217.8 Wh/m3 in Table 3. Please clarify.

Fixed

  1. The conclusion does not explain any correlation between the electrode material, properties and the CDI performance observed. There is very minimal in terms to addition to scientific body of knowledge in the area of capacitive deionization.

We agree with the reviewer’s suggestion. The conclusion was modified as follow:

In this research work, the effects of the thickness of the active material and flow rate in the performance of a CDI cell based on activated carbon electrodes were studied. In or-der to get the most efficient AC electrode, we fabricated and characterized electrodes with different thicknesses deposited on a graphite sheet. We found that electrodes with a thick-ness of 102 μm possess the best electrochemical properties and therefore were utilized to fabricate our CDI cell. The performance of the CDI cell was analyzed. We obtained a spe-cific absorption capacity (SAC) of 10.2 mg/g, and a specific energetic consumption (SEC) of 217.8 Wh/m3 at a flow rate of 55 ml/min. These results were contrasted with results avail-able in the literature. We can highlight that our CDI cell operating with a higher initial salt feed concentration results in higher performance.

  1. There are two Figures number 6. Please review the whole manuscript and rearrange.

Fixed

  1. References are not consistent. Please revise.

Fixed

Round 2

Reviewer 2 Report

Authors have made significant changes to the manuscripts as per the recommendations. Please review the reference format thoroughly and be consistent. e.g. issue numbers are given for some references while some of them just have volume numbers. Also terms like Neff should be properly presented (with "eff" subscripted)